Corrected: Author correction

# Hsp70 and Hsp40 inhibit an inter-domain interaction necessary for transcriptional activity in the androgen receptor

Bahareh Eftekharzadeh[1,2,7], Varuna C. Banduseela[3,7], Giulio Chiesa[1,2,7], Paula Martínez-Cristóbal[1,2,7], Jennifer N. Rauch[4], Samir R. Nath[3], Daniel M.C. Schwarz[4], Hao Shao[4], Marta Marin-Argany[1,2], Claudio Di Sanza[1,2], Elisa Giorgetti [3], Zhigang Yu[3], Roberta Pierattelli [5], Isabella C. Felli [5], Isabelle Brun-Heath[1,2], Jesús García [1], Ángel R. Nebreda[1,6], Jason E. Gestwicki [4], Andrew P. Lieberman[3] & Xavier Salvatella [1,2,6]

Molecular chaperones such as Hsp40 and Hsp70 hold the androgen receptor (AR) in an inactive conformation. They are released in the presence of androgens, enabling transactivation and causing the receptor to become aggregation-prone. Here we show that these molecular chaperones recognize a region of the AR N-terminal domain (NTD), including a FQNLF motif, that interacts with the AR ligand-binding domain (LBD) upon activation. This suggests that competition between molecular chaperones and the LBD for the FQNLF motif regulates AR activation. We also show that, while the free NTD oligomerizes, binding to Hsp70 increases its solubility. Stabilizing the NTD-Hsp70 interaction with small molecules reduces AR aggregation and promotes its degradation in cellular and mouse models of the neuromuscular disorder spinal bulbar muscular atrophy. These results help resolve the mechanisms by which molecular chaperones regulate the balance between AR aggregation, activation and quality control.

[1] Institute for Research in Biomedicine (IRB Barcelona), The Barcelona Institute of Science and Technology, Baldiri Reixac 10, 08028 Barcelona, Spain. [2] Joint BSC-IRB Research Programme in Computational Biology, Baldiri Reixac 10, 08028 Barcelona, Spain. [3] Department of Pathology, University of Michigan, Ann Arbor, MI 48109, USA. [4] University of California at San Francisco, Department of Pharmaceutical Chemistry, 675 Nelson Rising Lane, San Francisco, CA 94158, USA. [5] CERM and Department of Chemistry "Ugo Schiff", University of Florence, Via Luigi Sacconi 6, 50019 Sesto Fiorentino, Florence, Italy. [6] ICREA, Passeig Lluís Companys 23, 08010 Barcelona, Spain. [7] These authors contributed equally: Bahareh Eftekharzadeh, Varuna C. Banduseela, Giulio Chiesa, Paula Martínez-Cristóbal. Correspondence and requests for materials should be addressed to J.E.G. (email: jason.gestwicki@ucsf.edu) or to A.P.L. (email: liebermn@umich.edu) or to X.S. (email: xavier.salvatella@irbbarcelona.org)

Molecular chaperones are key components of the protein quality control system. In addition to their well-known roles in protein folding and degradation[1,2], they also stabilize specific conformations of proteins to enable the fast and efficient relay of signaling information. This aspect of their function is exemplified by their activity on steroid hormone receptors (SHRs), such as the glucocorticoid receptor (GR) and the androgen receptor (AR). It is known that a complex of heat shock proteins, including Hsp90, Hsp70, HOP, and Hsp40, is required to stabilize an inactive SHR conformation[3,4]. This activity is termed a holdase function because the apo-receptor is held in a structural state that is stable, soluble and prepared for binding to ligands[5]. Hormone binding then shifts the SHR into a distinct conformer of lower solubility that gains affinity for DNA and is capable of activating transcription. Intriguingly, at least some of the members of the chaperone machinery become more transiently associated with the SHR during this process[6], suggesting that these complexes are dynamic and play a role in stabilizing the conformation of the active receptor. In addition, a subset of the chaperones, including Hsp70, is also required for SHR turnover, through recruitment of E3 ubiquitin ligases[7]. Thus, the SHR system provides an excellent model for understanding how molecular chaperones balance the needs of protein folding, stability, function and quality control.

Pioneering studies by Pratt et al. have revealed the order of chaperone binding and release from SHRs[3,4] and defined the importance of their individual functions in cell-free systems and in cells[8]. In support of this notion, a recent study showed that Hsp70, HOP, and Hsp90 bind to the ligand-binding domain (LBD) of GR[9], holding it in a stable, open conformation. However, it is unknown whether this mechanism is conserved in other nuclear receptors; indeed, the Buchner group recently found that different Hsp90 co-chaperones are required for activating GR when compared to AR[10]. In addition the structural roles of the chaperones during SHR activation and turnover are also not clear: for example, it has been proposed that Hsp70 interactions with the LBD are involved in receptor triage[11,12], but chemical modulators of Hsp70 or Hsp40 induce ubiquitin-dependent degradation of AR variants that lack the LBD[13]. These observations suggest that some chaperones, especially Hsp70 and Hsp40, may have additional binding sites outside the canonical LBD region and that these might be critical for homeostasis of some SHRs.

A mechanistic knowledge of how SHRs are regulated by chaperones is also important for better understanding their roles in disease. For example, AR is an important therapeutic target for prostate cancer, because it controls the expression of genes associated with proliferation[14,15] in response to androgens, such as dihydrotestosterone (DHT)[16]. AR is also clinically important in spinal and bulbar muscular atrophy (SBMA), a progressive degenerative disorder of the neuromuscular system caused by the pathologic expansion of a polyglutamine (polyQ) tract in the disordered, N-terminal domain of AR[17]. In vitro and in cells, polyQ repeats beyond 37 residues in length cause hormone-dependent AR misfolding and aggregation. In patients and animal models, this misfolding manifests through the formation of nuclear inclusions in motor neurons and skeletal muscle cells that contain ubiquitinated and misfolded polyQ AR[18,19]. Importantly, the chaperone-bound form of polyQ AR appears to be protected from aggregation, such that it only becomes insoluble after androgen-mediated release of the chaperones. Together, these findings in prostate cancer and SBMA have created particular interest in understanding the molecular mechanisms of how chaperones control the solubility, activation and turnover of AR[8,12,20,21]. Such discoveries could lead to new therapeutics: for example, it has recently been shown that chemical activators of Hsp70 regulate the transcriptional activity of AR[22,23].

AR is a 919-residue protein composed of three domains: the N-terminal transactivation domain (NTD; residues 1–558), the central DNA-binding domain (DBD; 559-619) and the C-terminal LBD (669–919). The structures of the DBD and LBD bound to their ligands have been characterized by X-ray crystallography[24–27]. The NTD is intrinsically disordered[28], but solution nuclear magnetic resonance (NMR) spectroscopy has shown that it is rich in secondary structure, especially in regions key for transactivation[29,30]. The NTD of AR is one of the longest in the human SHR family and it includes the polyQ tract, starting at residue 58, which becomes expanded in SBMA[17]. Binding of androgens to the LBD causes a large conformational change that leads to the nuclear localization of the receptor, driven by exposure of a nuclear localization signal between the DBD and the LBD[16]. Despite its relevance for both transcriptional activity and the onset of SBMA, the nature of this conformational change is not yet well-established. Nonetheless, it is known that androgen binding exposes a hydrophobic patch in the LBD, termed activation function 2 (AF-2), that has high affinity for a 5-residue FQNLF motif in the N-terminal segment of the NTD[31]. The inter-domain interaction between this highly conserved motif and AF-2 has been characterized by various biophysical methods, including X-ray crystallography. These studies show that FQNLF binds to the LBD as an α-helix[32], but it is not clear why this conformational transition is also associated with the disassembly of the chaperone complex. It has been proposed that, similar to GR[9], molecular chaperones bind to the LBD of AR and that their affinities for it decrease upon interaction with DHT, but the mechanisms underlying this regulation are unknown.

To investigate the earliest stages of AR activation by androgens, we analyzed the interaction of Hsp70 and Hsp40 with the NTD of AR in vitro, by solution NMR[33], and in cells. Solution NMR is particularly well-suited to the characterization of transient protein–protein interactions (PPIs) involving disordered domains[34]. We found that Hsp70 and Hsp40 bind the NTD, with μM affinity and, strikingly, that the binding site contains the same FQNLF motif that interacts with AF-2 in the LBD during activation. This observation suggests a tug-of-war model in which the FQNLF motif can be bound by either chaperone or the LBD, such that hormone binding shifts partitioning between these two possibilities. In addition, we found that this part of the NTD has the propensity to form oligomers that are stabilized by interactions involving residues of the FQNLF motif, leading to aggregation; however its interaction with Hsp70 keeps the protein in solution, explaining why chaperones are so well positioned to block the aggregation and misfolding of AR. Given that Hsp40, Hsp70, and other chaperones bind near polyQ repeats in other proteins such as huntingtin this suggests a general mechanism[35,36] by which these molecular chaperones keep proteins harboring polyQ tracts soluble. Inspired by these findings, we used small molecules to stabilize the interactions of Hsp70 with AR containing an expanded polyQ tract in cellular and mouse models of SBMA. Consistent with the results of the biophysical studies, these treatments reduce aggregation and enhance clearance of polyQ-expanded AR. Our results thus provide a structural basis for how chaperones manage the stability, activation, solubility and turnover of AR, which should inform efforts to treat both prostate cancer and SBMA.

## Results

**Hsp70 and Hsp40 bind to the FQNLF motif in the AR NTD.** We used solution NMR to determine whether Hsp40 and Hsp70 bind to the NTD of AR and, if so, where this interaction occurs.

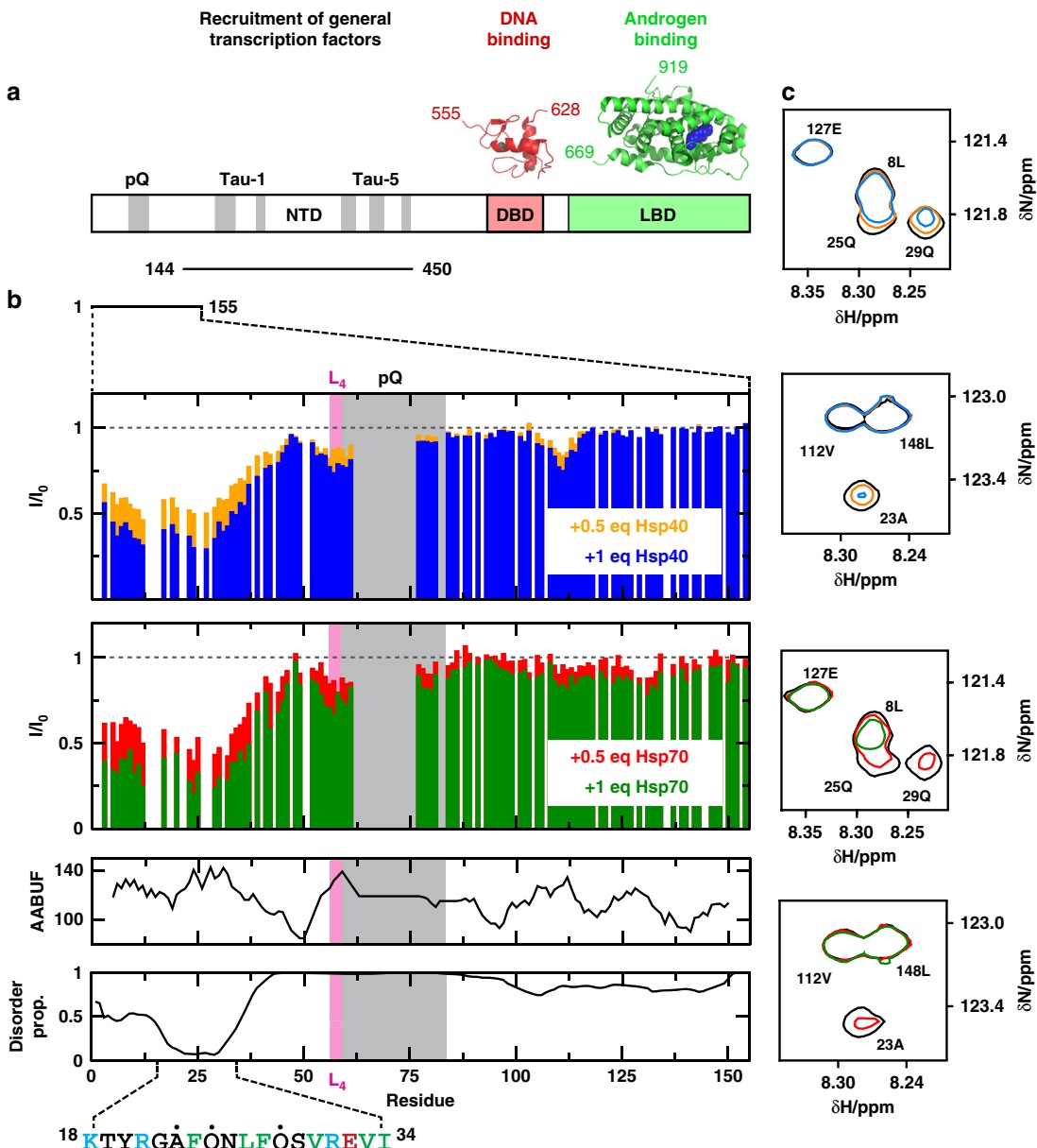

**Fig. 1** Hsp40 and Hsp70 interact with a region of the NTD containing the FQNLF motif. **a** Domain organization of the androgen receptor with an indication, in gray, of the regions of sequence of the transactivation domain (NTD) that are partially folded, of the structures of the globular domains (DBD and LBD) and of the two constructs used in this work (NTD$_{1-155}$ and NTD$_{144-450}$)[26,29]. **b** Changes in the intensity of the resonances of 33 μM NTD$_{1-155}$ caused by 0.5 (orange for Hsp40, red for Hsp70) and 1 (blue for Hsp40, green for Hsp70) molar equivalents of Hsp40 (top) and Hsp70 (bottom), in the absence of added nucleotides, with an indication of the accessible area buried upon folding (AABUF)[42], of the disorder propensity as predicted by PONDR[69], and of the interaction motif identified in this work, where positively charged residues are shown in blue, hydrophobic residues are shown in green and the residues labeled with a black dot are shown in **c**. **c** Selected regions of the $^1$H,$^{15}$N-HSQC spectrum of 33 μM NTD$_{1-155}$ in the presence of 0.5 and 1 molar equivalents of Hsp40 (top) and Hsp70 (bottom), in the absence of added nucleotides, that illustrate that no chemical shift changes are observed upon interaction with Hsp40 and Hsp70. In **c** only the contour of lowest intensity is shown for clarity, colored according to the code used in **b**

This study took advantage of recent reports that characterized two NTD constructs (NTD$_{1-155}$ and NTD$_{144-450}$) in solution[26,29]. Briefly, NTD$_{1-155}$ contains the FQNLF motif and the polymorphic polyQ repeat associated with SBMA, and spans residues 1–155 according to numbering used in the AR entry in the Uniprot database (http://www.uniprot.org/uniprot/P10275); the version of NTD$_{1-155}$ that we used contains 25 residues in the polyQ repeat, unlike the sequence found in Uniprot, that contains 23 (Fig. 1a). NTD$_{144-450}$ contains the regions of the NTD involved in interactions with transcription factors and transcriptional co-regulators, termed activation function 1 (AF-1)[29,30]. We incubated

NTD$_{1-155}$ (33 μM) or NTD$_{144-450}$ (33 μM) with either 0.5 or 1 molar equivalents of human Hsp70 (HSPA1A) or Hsp40 (DNAJA2). These experiments were performed in the absence of added nucleotides and samples were held at 5 °C, a temperature that maximizes spectral quality. The $^1$H,$^{15}$N-HSQC spectrum of NTD$_{144-450}$ was not altered by the addition of either Hsp40 or Hsp70, indicating weak or no interaction (Supplementary Fig. 1). This result is important because NTD$_{144-450}$ includes hydrophobic motifs that could represent non-specific sites for chaperone binding, yet none were observed to interact under these conditions. By contrast, some signals in the spectrum of NTD$_{1-155}$

decreased in intensity ($I/I_0 \sim 0.3$ at $[\text{Hsp}]/[\text{NTD}_{1-155}] = 1$) without changing their position in the spectra (Fig. 1b, c). This result indicates that the chaperones interact selectively with a region within $\text{NTD}_{1-155}$. We found that the states populated under these conditions - the free state and, likely, various bound states[37] - are in intermediate-to-slow exchange on the NMR timescale. The low temperature used to carry out these experiments (5 °C) might contribute to the low observed exchange rate. Also this scenario is typical for chaperone binding[38,39] and suggests a dissociation constant lower than the concentration used in the experiments i.e. in the low micro molar range (see below).

In these experiments, clear decreases in signal intensities were observed in the first 35 residues of $\text{NTD}_{1-155}$, which, strikingly, includes the motif FQNLF; this motif forms a complex with the LBD after activation by androgens[31,32]. We observed similar changes for Hsp70 and Hsp40, which is consistent with the known similarity of consensus sites for these two chaperones (see below)[40,41]. In addition, we observed much smaller decreases in signal intensity in other residues of $\text{NTD}_{1-155}$ likely caused by transient interactions. For example, intensity changes were observed in the hydrophobic $^{55}\text{LLLL}^{58}$ motif that induces a helical structure in the polyQ tract of AR[26,27]. In summary we found that, prior to activation by androgens, Hsp70 and Hsp40 bind to the NTD of AR by selectively interacting with a short region that includes the FQNLF motif.

To determine whether this motif is an important part of the region that binds to Hsp40 and Hsp70, we produced a mutant $\text{NTD}_{1-155}$, named $\text{AQNAA-NTD}_{1-155}$, in which residues Phe 24, Leu 27 and Phe 28 were mutated to Ala. These mutations were predicted to decrease the hydrophobicity of the putative binding site, expressed as AABUF[42] (Supplementary Fig. 2A), a property that is key for the ability of these molecular chaperones to recognize their substrates[37,43]. By NMR, neither Hsp40 nor Hsp70 interacted with $\text{AQNAA-NTD}_{1-155}$, confirming that the FQNLF motif is indeed important for the interaction between these molecular chaperones and this region of AR (Supplementary Fig. 2A).

There are two possible regions of Hsp70 where substrates bind to: a canonical site in a hydrophobic groove in the substrate binding domain (SBD) and a non-canonical interaction surface[44]. To test which site was used by AR's $\text{NTD}_{1-155}$ we titrated it into solutions of $\text{Hsp70}_{\text{SBD}}$ (residues 394 to 509) and measured competition for a model, canonical peptide (LVEALY-FAM) by fluorescence polarization (FP) in the absence of nucleotides. We found that $\text{NTD}_{1-155}$ competed with LVEALY-FAM (Fig. 2), even better than the NR peptide, a bona fide Hsp70 substrate, suggesting that it bound with an $\text{IC}_{50}$ $1.7 \pm 0.5\,\mu\text{M}$ to the canonical binding site. We repeated these experiments by using the $\text{AQNAA-NTD}_{1-155}$ mutant and found, in agreement with the NMR experiments, that it had ~10-fold weaker affinity for Hsp70 (Supplementary Fig. 3). To further verify this interaction by an independent method, we measured the affinity between $\text{NTD}_{1-155}$ and Hsp70 by isothermal titration calorimetry (ITC). Although it was challenging to quantify the affinity because of solubility limitations, the binding affinity was estimated to be ~5 μM (Fig. 2b). This value is in good agreement with the FP studies as well as with the NMR titrations because this dissociation constant should lead, in the slow exchange regime, to $I/I_0 \sim 0.3$ when the $\text{NTD}_{1-155}$ (at 33 μM) is in the presence of an equimolar amount of molecular chaperone (see Fig. 1b). Together, these results suggest that the region of sequence corresponding to the FQNLF motif of AR and its flanking regions represents a canonical substrate for Hsp70.

As mentioned above, Hsp40 and Hsp70 often compete for the same sites on their client proteins. To understand the relative

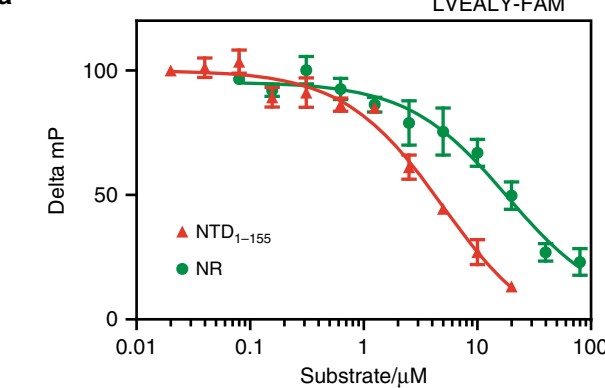

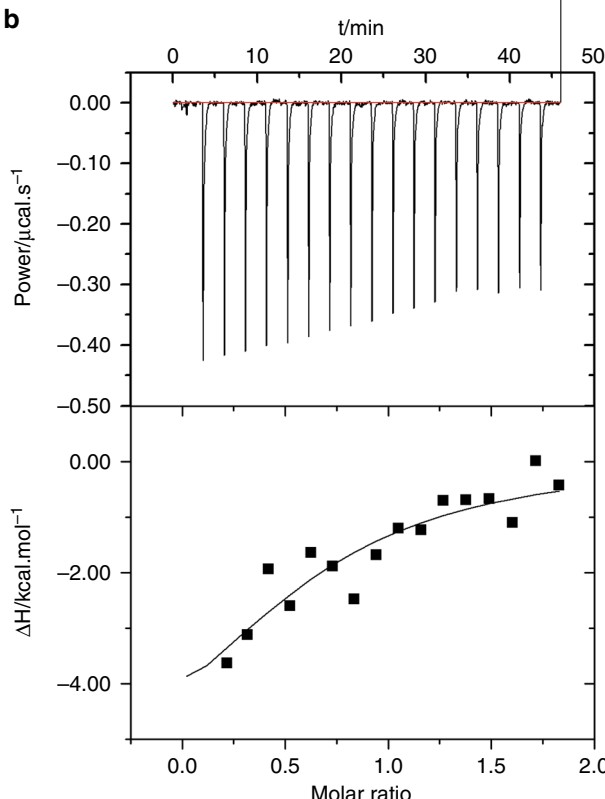

**Fig. 2** The FQNLF motif binds to the canonical binding site in Hsp70 in the absence of nucleotides. **a** Binding of $\text{NTD}_{1-155}$ to the canonical cleft of $\text{Hsp70}_{\text{SBD}}$ at 2 μM in the absence of nucleotides was measured by competition with the canonical peptide LVEAVY labelled with a FAM fluorophore (25 nM) as measured by FP. NR, a bona fide substrate of $\text{Hsp70}_{\text{SBD}}$, was used as positive control. Data are the average of at least three independent experiments performed in duplicate on different days and the error bars represent S.E.M. **b** Direct binding of $\text{NTD}_{1-155}$ to $\text{Hsp70}_{\text{SBD}}$ in the absence of nucleotides was measured by ITC: the top panel represents the raw data whereas the lower panel represents the integrated heat peaks and the fit to the one-site binding model. Source data for **a** are available as Source Data file

affinities of these chaperones, we measured binding of purified, human Hsp40 (DnaJA2) for $\text{NTD}_{1-155}$ by ITC. Hsp40 bound slightly tighter to $\text{NTD}_{1-155}$, with a Kd of $3.3 \pm 2.3\,\mu\text{M}$ (Supplementary Fig. 3B). The stoichiometry of the complex was estimated to be ~2 Hsp40s per $\text{NTD}_{1-155}$, ($n = 0.49 \pm 0.27$)

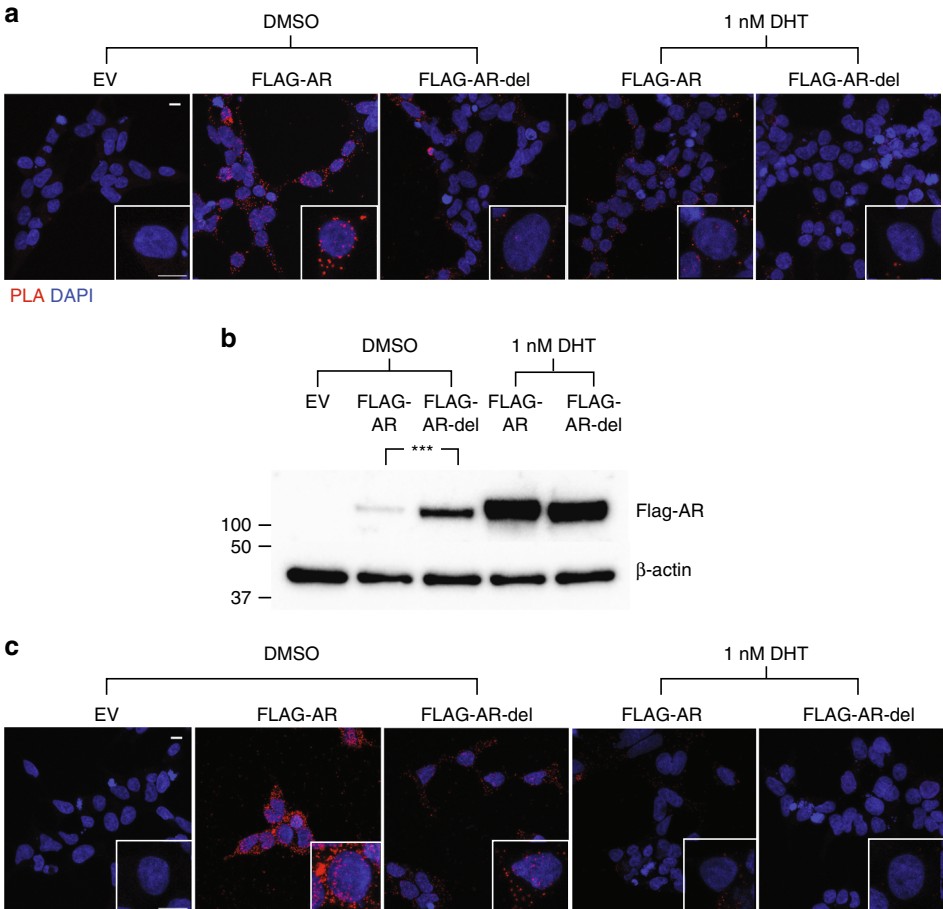

**Fig. 3** Removal of the FQNLF motif and flanking residues prevents the interaction of cytosolic AR with Hsp40 and Hsp70 in cells. **a** The interaction between FLAG-AR as well as FLAG-AR-del with endogenous Hsp70 in the absence or presence of 1 nM DHT was studied in transfected HEK293T cells by PLA ($n =$ 3). EV stands for empty vector, FLAG-AR/Hsp70 complexes appear as red dots and DAPI (blue) indicates the localization of nuclei. **b** Analysis by Western blotting of the levels of FLAG-AR and FLAG-AR-del in transfected HEK293T cells in the absence or presence of 1 nM DHT ($n = 3$). β-actin immunoblot is shown as a loading control. Log2 transformed values were used to assess differences using two-tailed unpaired $t$-tests and ***$p < 0.001$. **c** Assay equivalent to that presented in **a** to study the interaction with the endogenous Hsp40 ($n = 3$). The scale bars in **a**, **c** represent 10 μm. Source data for the comparison of protein levels shown in **b** are available as Source Data file

consistent with the dimeric structure of this chaperone. These results suggest that both Hsp70 and Hsp40 bind with low micromolar affinity to $NTD_{1-155}$.

**Hsp70 and Hsp40 bind to the FQNLF motif in cells.** To validate in cells the observations made in vitro HEK293T cells were transfected with full length AR tagged with the FLAG epitope (FLAG-AR) and its interaction with endogenous Hsp40 and Hsp70 was assessed by proximity ligation assay (PLA). This immunofluorescence-based technique is particularly useful for detecting transient PPIs in cells where the putative complexes are of intermediate or low stability and therefore unlikely to withstand the washing steps that are required for co-immunoprecipitation experiments[45]. Briefly, the method relies on the use of primary antibodies that recognize the two proteins (e.g. AR and Hsp70), and on secondary antibodies that are conjugated to complementary oligonucleotides. When the proteins are in close proximity, these oligonucleotides hybridize and are ligated to form a circular DNA template that can be amplified by DNA polymerase and detected by fluorescence probes. The PLA signals can therefore be detected by fluorescence microscopy as red puncta in the PLA experiment.

Using this approach, we observed a clear interaction between FLAG-AR and both Hsp40 and Hsp70 in the cytoplasm prior to activation but found that DHT treatment induced an almost complete loss of the interaction (Fig. 3a, c) combined with nuclear translocation (Supplementary Fig. 4). These results agree with the notion that activation of AR by DHT causes the dissociation of the cytosolic complexes that AR forms with molecular chaperones prior to nuclear translocation[3,8]. To test whether the interaction detected by PLA occurs in cells via the region of sequence containing the FQNLF motif detected by NMR, we repeated these experiments using a version of FLAG-AR with the FQNLF motif mutated to AQNAA (FLAG-AR-mut) and a deletion mutant, where the region of sequence spanning residues 21 to 35, which includes the FQNLF as well as adjacent similar VREVI motif, was deleted (FLAG-AR-del). We observed that while FLAG-AR-mut could still interact with both Hsp40 and Hsp70 (Supplementary Fig. 5), the interaction of FLAG-AR-del with both molecular chaperones was substantially weaker (Fig. 3a, c). These results suggest that mutation of the three residues at the core of the FQNLF motif is not sufficient to abolish the affinity for Hsp70 or Hsp40, that instead requires the complete deletion of this motif and the ca. 10 residues immediately flanking it. This result is consistent with the 10-fold weakened affinity of the AQNAA

mutant for Hsp70 observed in the NMR and FP experiments (see Supplementary Fig. 3). It should be noted that our PLA experiments were designed to detect the interaction of molecular chaperones with the NTD of AR and are therefore unlikely to report on whether they interact with the LBD. In addition, the fact that both FLAG-AR-mut and FLAG-AR-del readily translocate to the nucleus upon activation suggests that these variants retain some early AR functions (Supplementary Fig. 4). Collectively, these results confirm that, in cells, prior to activation by androgens, the molecular chaperones Hsp40 and Hsp70 bind to AR via a region of sequence including the motif FQNLF and the residues flanking it.

Hsp70 and Hsp40 have both been implicated in turnover of AR. To probe the role of their interaction with the FQNLF motif in this process, we compared the levels of FLAG-AR-mut and FLAG-AR-del with those of the wild type protein prior and following to activation with DHT. We observed that mutation and, especially, deletion of the FQNLF motif caused an increase in basal AR levels in the absence of hormone (Fig. 3b, Supplementary Fig 5B). Also, recent studies have shown that disrupting AR's N/C interaction delays disease onset in a mouse model of SBMA[46]. These studies utilized mice that expressed a mutant form of the polyQ AR where the Phe residue of the FQNLF motif was mutated to Ala. To determine whether this mutation had an effect on the interaction between the AR NTD and molecular chaperones, we carried out experiments with the equivalent mutant of $NTD_{1-155}$, named $AQNLF-NTD_{1-155}$. An analysis of the interaction by NMR showed that the AQNLF mutant interacts in vitro with molecular chaperones with an affinity that was indistinguishable from that of the WT (Supplementary Fig. 6). Thus, our results confirm that amelioration of the SBMA phenotype in mice expressing this mutant is likely due to a weakening of the N/C interaction rather than an altered interaction with molecular chaperones.

**Hsp70 increases the solubility of the AR NTD.** Because Hsp70 and Hsp40 are released from the $NTD_{1-155}$ after addition of the androgen dihydrotestosterone (DHT), the hormone-dependent phenotype of SBMA and ligand-dependent aggregation of the polyQ AR strongly suggest a protective role for these molecular chaperones in limiting AR aggregation[47–50]. To investigate this possibility by solution NMR, we analyzed the oligomerization properties of the $NTD_{1-155}$ construct by measuring $^1H,^{15}N$-HSQC spectra at two different concentrations, 125 and 450 μM at 278 K. NMR can be used to monitor concentration-dependent oligomerization because the monomer-to-oligomer exchange process contributes to NMR linewidths, such that the $R_2$ $^{15}N$ relaxation rates are increased[51]. In $NTD_{1-155}$ we first confirmed that the position of the backbone amide chemical shifts did not change with concentration. Then, we measured the $R_2$ $^{15}N$ relaxation rates and found that some residues had higher values at 450 μM, when compared to 125 μM (Fig. 4a). A closer examination of these linewidth changes showed that the affected residues were clustered into regions of locally high AABUF values, suggesting that hydrophobicity could be the main driving force of oligomerization. Indeed, the most affected region was localized to the first 50 residues of the construct, with a maximum in the FQNLF motif. This observation is conceptually equivalent to that made by Wetzel and co-workers in huntingtin, in which it was shown that the early stages of aggregation are not driven by inter-molecular interactions involving the polyQ tract, but rather by hydrophobic, coiled coil-like interactions in the sequence flanking it at the N-terminus[52,53]. Further, these results raise the intriguing possibility that binding of molecular chaperones to this region of AR might directly suppress its oligomerization.

We then carried out NMR experiments to directly study whether the interaction of Hsp70 with the $NTD_{1-155}$ construct influenced its solubility. For these experiments, we prepared samples of $NTD_{1-155}$ at 125 μM and added human Hsp70 at 1 molar equivalent, using the same conditions presented in Fig. 1, and measured the corresponding $^1H,^{15}N$-HSQC spectra as a function of incubation time at 298 K. In the absence of chaperone, we observed a progressive decay of signal intensity, so that after 6 days of incubation, ~25 % of the signal intensity was lost. This signal decay was likely caused by formation of aggregates whose size precluded their detection by NMR, as evident by the turbidity of the samples over time. By contrast, the samples containing Hsp70 did not show any decrease of signal intensity, suggesting that the complex remained soluble (Fig. 4b). These results suggest that molecular chaperones limit self-interactions between FQNLF motifs in different AR NTD molecules, diminishing its oligomerization and maintaining solubility.

**Hsp70 increases AR solubility in vivo.** A key prediction of these studies is that chemical stabilization of the Hsp70 interaction might limit aggregation of polyQ AR. Recent work has produced chemical probes that significantly enhance Hsp70's affinity for disordered proteins[54]. Briefly, Hsp70 normally cycles through rounds of ATP hydrolysis, with the ATP-bound state having relatively poor affinity for substrate proteins and the ADP-bound state having tight affinity[55]. These compounds, such as YM-1, favor the ADP-bound state[56], therefore stabilizing Hsp70's interactions. In addition, stabilizing this complex has been shown to enhance turnover of some Hsp70 client proteins[54]. Thus, we hypothesized that these compounds might limit polyQ AR aggregation and potentially promote its turnover in SBMA models.

To understand if Hsp70 modulators might have these favorable activities, we first tested a small panel of them in PC12 cells that stably express AR with an expanded polyQ tract encoded by 112 CAG repeats, termed AR112Q, under a tetracycline responsive promoter[57]. In these cells, treatment with ligands, such as DHT or the synthetic androgen metribolone (R1881), leads to nuclear accumulation of misfolded polyQ AR, consistent with aggregation after release of chaperones. SDS-insoluble polyQ-AR species can be separated by centrifugation and visualized by western blot or by filter trap assay (Supplementary Fig. 7). In our small-scale screen, we found that two Hsp70 modulators, JG-84 and JG-98, were the most potent in enhancing solubility of polyQ AR (Supplementary Fig. 7A-D). Moreover, the active molecules promoted clearance of both monomeric and aggregated polyQ AR species (Supplementary Figs. 7D–F, 8), and were more striking on polyQ AR than wild type AR (Supplementary Fig. 8). Further analysis demonstrated that the more potent Hsp70 modulator, JG-98, increased polyQ AR ubiquitination and enhanced degradation through the proteasome (Fig. 5a, b). Notably, treatment with JG-258, a structural analog of JG-98 that does not bind Hsp70[58], does not change steady state polyQ AR levels or alter polyQ AR ubiquitination (Supplementary Fig. 9).

We wondered whether these effects might occur because of an HSF1 dependent stress response. To test this idea, we performed western blots for three stress-inducible biomarkers: Hsp25, Hsp40, and Hsp70. In treated cells, reduction of AR112Q occurred without induction of heat shock proteins (Fig. 5c). In contrast, the Hsp90 inhibitor, 17-AAG, strongly increased the levels of these proteins (Fig. 5c), consistent with induction of an HSF1 dependent stress response. Additionally, JG-98 did not affect levels of Hsp90 client proteins in their native conformations, such as Akt and Erk (Fig. 5c), both of which are degraded following treatment with Hsp90 inhibitors. Finally, recent work

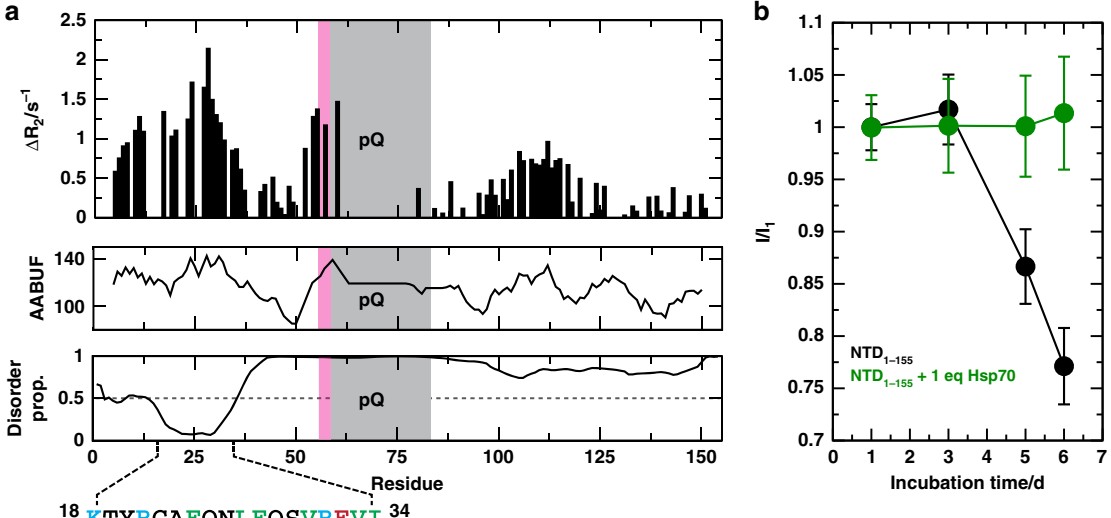

**Fig. 4** Hsp70 prevents aggregation of the AR NTD. **a** Plot of the difference in $^{15}$N $R_2$ relaxation rates of the NMR resonances of the NTD$_{1-155}$ construct at 125 and 450 µM and of the AABUF[42] and disorder propensity, as predicted by PONDR[69], of the NTD$_{1-155}$, indicating the presence of NTD$_{1-155}$ oligomers stabilized by hydrophobic interactions. **b** Decrease in the average intensity of the resonances of the NTD$_{1-155}$ construct as a function of incubation time at 298 K in the absence and in the presence of 1 molar equivalent of Hsp70, indicating that this molecular chaperone abolishes the aggregation of NTD$_{1-155}$ at 125 µM at 278 K and in the absence of nucleotides. Data are mean ± S.D. from the values obtained for all residues whose intensities are not affected by Hsp70 binding. Source data for panel **b** are available as Source Data file

has suggested that combining Hsp70 modulators with Hsp90 inhibitors might further promote AR clearance[12]. Consistent with this notion, we found that combinations of an Hsp70 modulator, JG-98, with an Hsp90 inhibitor, 17-AAG, produced robust polyQ AR clearance (Fig. 5d). Together, these findings supported a model in which stabilizing the Hsp70-AR interaction enhances the solubility and clearance of AR.

To test this idea in an animal model of SBMA, we turned to an analog of the Hsp70 modulators, JG-294, that is tolerated in mice and has well-studied pharmacokinetics[58]. Before proceeding into animals, we first confirmed that JG-294 could induce degradation of polyQ AR in PC12 cells expressing AR112Q. Indeed, treatment with JG-294 yielded a significant reduction in the levels of AR112Q, as detected by western blot and immunofluorescence microscopy (Fig. 6b, c). Next, we tested whether JG-294 might have activity in a well-characterized mouse model of SBMA generated by gene targeting (AR113Q mice)[50,59]. In this model, like the cell-based system, AR113Q accumulates as SDS-insoluble high molecular weight species in skeletal muscle from mutant males. Strikingly, treatment with JG-294 for two weeks (i.p., at 3 mg kg$^{-1}$ every other day) reduced levels of both high molecular weight and monomeric polyQ AR when compared to the vehicle treated controls (Fig. 7a, b, d, e). These changes occurred without alteration in AR mRNA levels (Fig. 7c), consistent with the notion that JG-294 promoted polyQ-AR clearance. Notably, as in cell culture, these effects were distinct from those triggered by treatment with an Hsp90 inhibitor, as no induction of Hsp70 or loss of client proteins in their native conformation were seen following treatment with JG-294 (Fig. 7e). Together, these results suggest that stabilizing the interactions between Hsp70 and AR may be a viable therapeutic strategy for SBMA. Collectively, our findings support a model in which Hsp70 directly binds to the NTD of AR as a sensor for misfolding and aggregation.

## Discussion

We show here that AR activation seems to be governed, in part, by a competition between chaperones and the AF-2 region for binding to a region of the NTD including the FQNLF motif.

Specifically, inter-molecular interactions with Hsp70 and Hsp40 appear to prevent activation of AR by sequestering the FQNLF motif while intra-molecular interactions of this same region with AF-2 in the LBD are required for activation. We find that the relative strength of these competing PPIs is regulated by androgen binding to the LBD, which stabilizes the AF-2, and, therefore, promotes its interaction with the FQNLF motif and displaces the chaperones.

What is the role of the Hsp40 in this process? Hsp40 and Hsp70 normally work in concert and, by analogy with other systems, we propose that Hsp40 binds AR first and then recruits Hsp70 via its conserved J-domain. In cells, binding of the J-domain stimulates ATP cycling by Hsp70[60], which creates the tight affinity, client-chaperone complex. Therefore, we expect that Hsp40 would assist Hsp70 by both localizing it to AR and accelerating its catalytic cycle. This is important because our in vitro experimental platforms, while aimed at investigating the equilibrium properties of the system in the absence of nucleotide, likely underestimate the effective affinity for the NTD. Moreover, there are more than 45 proteins in the Hsp40 family in humans, so the relevant subset for AR homeostasis is not yet clear.

One of the interesting implications of this model is that the structure of the FQNLF motif itself might switch when bound to either the chaperones or AF-2. Specifically, it is known that the FQNLF motif is helical after binding to AF-2 and activation, yet we have shown that it can interact with the canonical groove of Hsp70, likely in an extended conformation. Thus, these studies suggest that (at least) two sets of competing PPIs, in addition to a local conformational switch, underlie the activation of AR. Further experiments will be needed to understand whether there is a link between these molecular events and the broader changes that occur during nuclear translocation and transcriptional activity of AR. Additionally, it has previously been observed that Hsp70 binds to the LBD of SHRs[9] and it is not yet known whether the N-terminal mechanism presented here is connected to other chaperone-mediated processes at the LBD. Despite these remaining questions, this molecular view of AR dynamics and PPIs has a number of interesting implications.

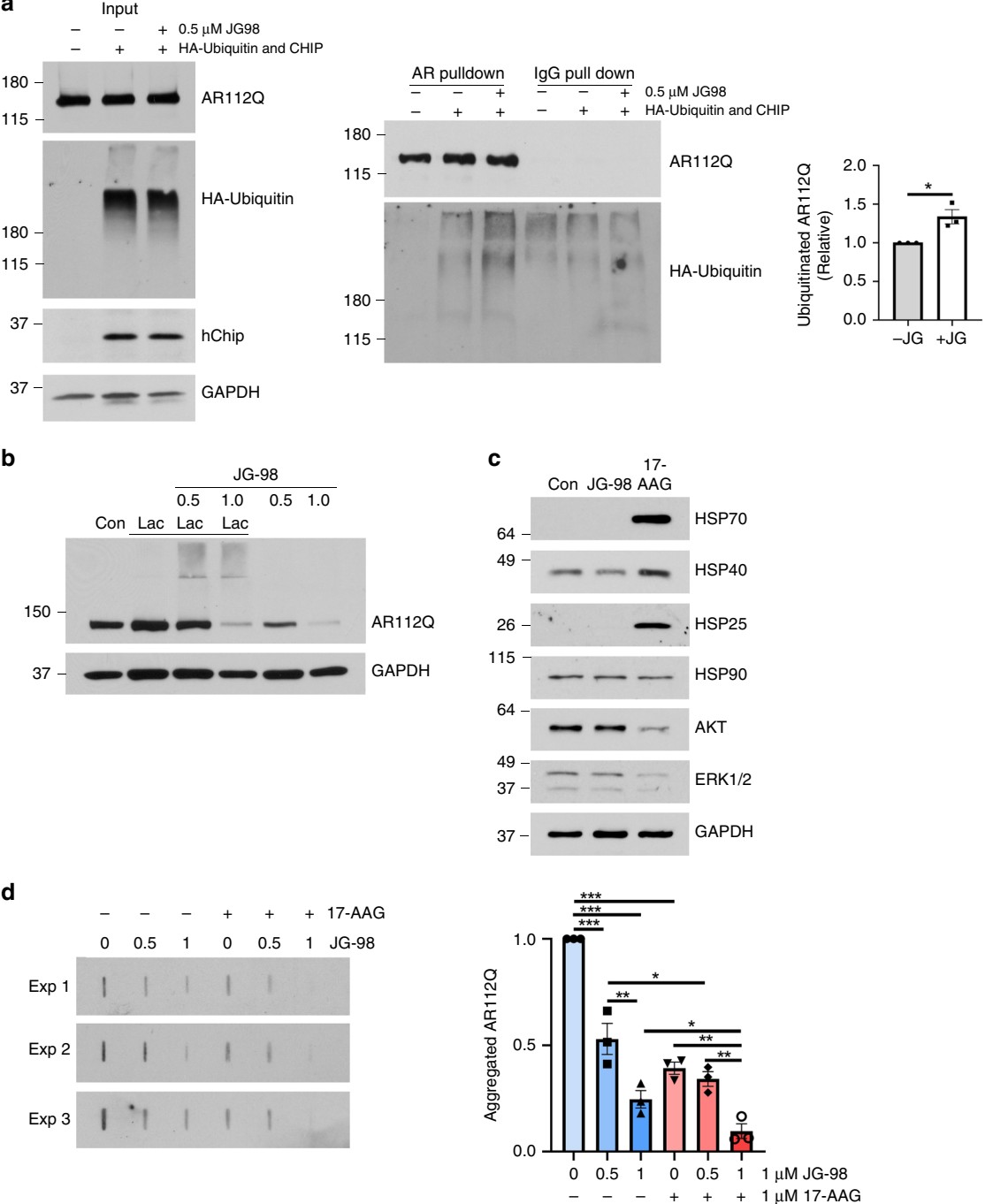

**Fig. 5** Hsp70-targeted small molecules increase ubiquitination and degradation of polyQ-AR. **a** PC12 cells were transiently transfected to express HA-ubiquitin and CHIP, and AR112Q expression was induced for 48 h. Cells were treated with the Hsp70 modulator, JG-98 (0.5 μM) for the last 24 h and with 10 μM MG132 for the last 16 h. AR112Q was immuno-precipitated from lysates and then probed for ubiquitin (HA). Left: input. Middle: pull down. Right: quantification from three independent experiments. Data are mean ± S.E.M. from three independent experiments. *$p < 0.05$ by two-tailed $t$-test. **b** JG-98 promotes AR112Q degradation by the proteasome. PC12 cells were induced to express AR112Q for 48 h in the presence of R1881 (10 nM) and JG-98 (0.5 or 1.0 μM). Indicated samples were treated with lactacystine (10 μM) for the final 16 h. AR was detected by western blot. GAPDH controls for loading. **c** JG-98 does not induce a stress response. PC12 cells were induced to express AR112Q in the presence of R1881 (10 nM) for 48 h in the presence of 17-AAG (5 μM), JG-98 (0.5 μM) or vehicle control. Cell lysates were probed for Hsp25, Hsp40, Hsp70, Hsp90, Akt, and ERK1/2. GAPDH controls for loading. **d** PC12 cells were induced to express AR112Q in the presence of R1881 for 48 h, and treated with 17-AAG (1 μM) and/or JG-98 (1 μM), as indicated. Lysates were analyzed for AR by filter trap assay. Data are mean ± S.E.M. from three independent experiments. *$p < 0.05$, **$p < 0.01$, ***$p < 0.001$ by one-way ANOVA. Source data for the quantifications shown in **a**, **d** are available as Source Data file

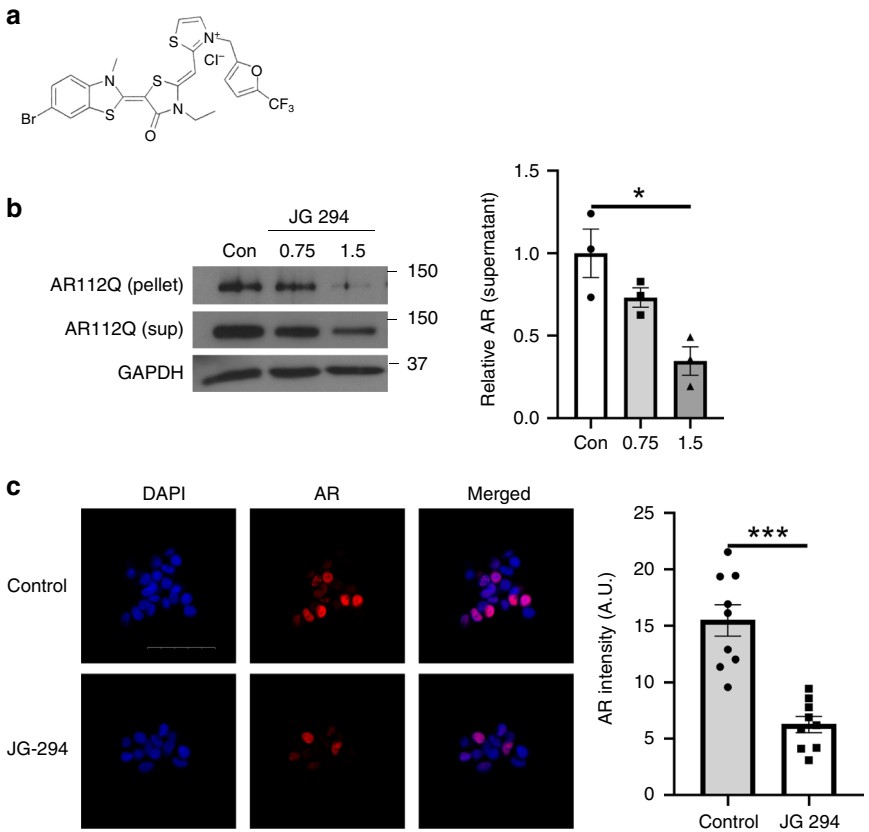

**Fig. 6** JG-294 promotes AR112Q clearance. **a** Chemical structure of JG 294. **b** PC12 cells were induced to express AR112Q in the presence of R1881 (10 nM) for 48 h, washed to remove doxycycline to turn off transgene expression, and then incubated for 48 hr in the presence of JG-294 (0.75, 1.5 μM) or vehicle control (Con). Left: AR levels were analyzed by western blot. GAPDH controls for loading. Right: quantification from 3 independent experiments. Data are mean ± S.E.M. *$p < 0.05$ by one-way ANOVA. **c** PC12 cells were induced as in panel **b**, and treated with JG-294 (1.5 μM) for 48 h. Left: Cells were fixed and stained for AR (red). Nuclei were stained with DAPI (blue). Scale bar, 50 μm. Right: Fluorescence intensity of nuclear AR quantified from three independent experiments. Data are mean ± S.E.M. ***$p < 0.001$ by two-tailed $t$-test. Source data for the quantifications shown in **b**, **c** are available as Source Data file

First, these results also provide insight into how Hsp70 and Hsp40 regulate the solubility of AR. We noted that Hsp70 prevents the oligomerization of AR's NTD$_{1-155}$ (Fig. 4), likely by directly encumbering the FNQLF motif and surrounding regions that are hydrophobic and aggregation-prone. We speculate that this activity, combined with the fast turnover of AR prior to androgen binding, conspire to minimize the propensity of the receptor to oligomerize and aggregate by preventing exposure of the most aggregation-prone sequence while simultaneously keeping its concentration low in the cytoplasm. In this scenario, molecular chaperones would contribute, by acting as holdases, to keeping the receptor monomeric and at low concentration i.e. primed for rapid activation by androgens without compromising solubility. Putting these mechanisms together, it seems elegant to propose that Hsp70's interactions with the FQNLF motif might create a three-way tug-of-war, in which the chaperone competes for both: (a) proper intra-molecular contacts with AF-2 and (b) inter-molecular contacts between the FQNLF motifs of two AR molecules, leading to aggregation. The relative affinities of these competing PPIs likely balances the activation of the receptor and governs its unnatural aggregation in SBMA (Fig. 8).

Based on the experiments with Hsp70 modulators, we propose that this delicate balance of PPIs also regulates AR turnover. In other systems, such as microtubule-associated protein tau (MAPT) and X-linked inhibitor of apoptosis (XIAP), tight binding of Hsp70 has been observed to promote turnover[54,61].

Similarly, prolonged interactions with Hsp70 seem to favor turnover of polyQ-expanded AR. Thus, more broadly, Hsp70 seems to use a dwell time method to discriminate between damaged and healthy proteins (Fig. 8). In the case of AR, the incorrect decision to release polyQ AR may partially underlie protein aggregation in SBMA, such that chemical perturbations with JG-98 or JG-294, can partially re-balance the system and promote turnover. In this model, it is striking that the same chaperone that governs receptor activation and solubility is also tasked with monitoring its quality control.

Notably, numerous functions essential for cell viability, ranging from energy and ion homeostasis to axonal transport and gene expression, are disrupted by misfolded proteins that cause age-dependent protein aggregation neurodegenerative disorders, including SBMA[62,63]. Alterations in these multiple pathways likely contribute to toxicity in a cumulative manner, suggesting that therapeutically targeting any one of them is likely to yield incomplete rescue. In contrast, decreasing levels of mutant proteins such as the polyQ AR has emerged as an attractive therapeutic strategy to target the proximal mediator of disease pathogenesis.

In this regard, our findings have implications for potential new therapies. Many efforts to develop chaperone-targeted therapies for protein aggregation diseases have focused on inhibition of Hsp90. Our data indicate that targeting Hsp70 may have significant advantages. For example, Hsp70 activation favors the

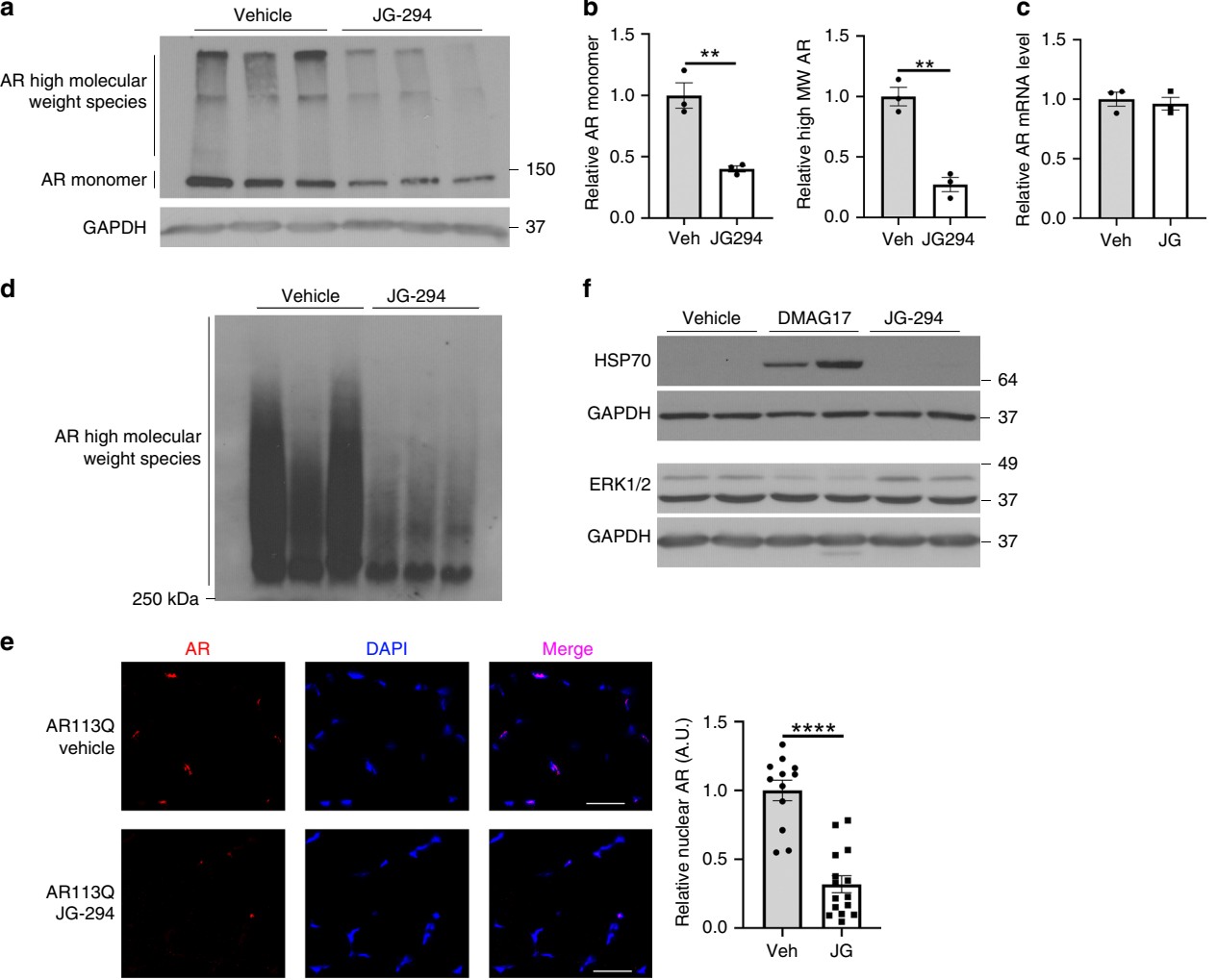

**Fig. 7** JG-294 promotes clearance of AR113Q in mouse skeletal muscle. Three-month-old AR113Q males were injected i.p. with JG-294 (3 mg kg$^{-1}$), 17-DMAG (10 mg kg$^{-1}$), or vehicle every other day for two weeks. **a** Protein lysates from quadriceps were analysed by western blot for AR and GAPDH. **b** Quantification of AR monomer and AR high molecular weight species by densitometry. Data are mean ± S.E.M. from three mice per group. **p < 0.01 by two-tailed $t$-test. V, Vehicle; JG, JG-294. **c** Relative expression of AR mRNA was determined by qPCR. Data are mean ± S.E.M. from three mice per group. **d** Quadriceps muscle protein lysates were analyzed by agarose gel electrophoresis for resolution of aggregates (AGERA) followed by western blot for AR. **e** Quadriceps were stained for AR (red) and analyzed by immunofluorescence microscopy. Nuclei were stained by DAPI (blue). Scale bar = 100 μm. At right, Quantification of AR signal within nuclei. Data are mean ± S.E.M. from three independent experiments. ***P < 0.0001 by two-tailed $t$-test. **f** Protein lysates from quadriceps were probed by western blot for Hsp70, Erk1/2 and GAPDH. Source data for the quantifications shown in panels **b**, **c**, and **e** are available as Source Data file

removal of misfolded proteins rather than client proteins in their native or near-native states[11,12,64]. This distinction suggests that Hsp70-targeted small molecules may avoid some of the toxic effects of Hsp90 inhibitors, which promote degradation of hundreds of client proteins in their native conformations, some of which might be needed for normal cellular functions. Similarly, the other major strategy for activating chaperone function is to target the stress response transcription factor, HSF1. However, experimental evidence suggests that chronic activation of HSF1 may promote toxicity by triggering a maladaptive stress response that is present in many protein-folding diseases[65]. In contrast, we found that targeting Hsp70 does not lead to activation of stress-responsive proteins. Thus, these results suggest that Hsp70 may be a good target for the treatment of SBMA. In addition, AR has been implicated in other diseases, such as prostate cancer. Indeed, recent studies have established that the same Hsp70-targeted small molecules promote clearance of an AR splice variant that is

associated with prostate cancer and lacks the C-terminal LBD[13]. Together, these observations point to Hsp70 as a promising target in multiple AR-associated diseases. Such strategies may take advantage of the normal protein quality control functions of this chaperone, which are already evolved to regulate AR solubility and degrade the damaged receptor.

## Methods

**Protein expression and purification**. The genes codifying for NTD$_{1-155}$ and its variants were purchased from GeneArt and cloned in a pDEST-HisMBP vector (Addgene: #11085) whereas that codifying for NTD$_{144-450}$[29] was instead cloned into a Gateway pDEST17 vector (Invitrogen). Expression of the resulting genes led to fusion proteins containing a His$_6$ tag, a maltose binding protein (MBP) moiety for NTD$_{1-155}$ and its variants and a motif recognized by the tobacco etch virus (TEV) protease. The His$_6$ tag was used as a purification tag, whereas MBP was used to increase the solubility of NTD$_{1-155}$ and its variants. $^{15}$N labeled protein expression was carried out in Rosetta (DE3) pLysS E. coli competent cells Novagen (Merck), grown in Luria Broth (LB; Melford) at 37 °C at 220 r.p.m. until the value of OD$_{600}$ was 0.7, centrifuged at 2000 G for 20 min, then washed in MOPS buffer (3-(N-

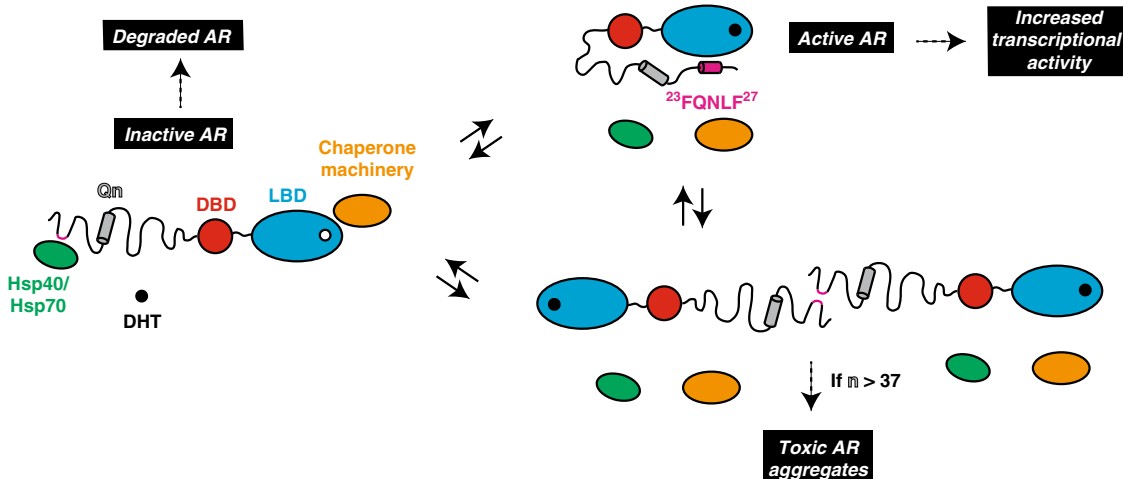

**Fig. 8** Roles of Hsp40 and Hsp70 in AR homeostasis. Scheme describing how Hsp40 and Hsp70, by interacting with a region of sequence containing the FQNLF motif, regulate the activity, cellular concentration and solubility of AR

morpholino) propanesulfonic acid (200 mM), sodium acetate (50 mM), EDTA (10 mM), pH 7.0) devoid of nitrogen sources and rapidly centrifuged at 5000 G for 10 minutes. Cells were then resuspended in MOPS media containing $^{15}NH_4Cl$ as a nitrogen source (Eurisotope), induced with 0.5 mM Isopropyl β-D-1-thiogalacto-pyranoside (IPTG, Sigma) for 4 hours at 28 °C, harvested by centrifugation and resuspended in core buffer (50 mM sodium phosphate, 500 mM NaCl, 5% (v/v) Glycerol, 1 mM 2-Mercapto-ethanol, pH 8.0). A HisTrap HP 5 ml column (GE Healthcare) was used to purify the proteins, which were eluted by an imidazole gradient (final composition: 500 mM imidazole, 50 mM sodium phosphate, 500 mM NaCl, 5% Glycerol, 1 mM βMercapto-ethanol, pH 8.0), followed by a size exclusion step carried out in a Superdex HighLoad S200 26/60 column (GE Healthcare) equilibrated in a buffer with the following composition: 500 mM NaCl, 12 mM sodium phosphate, 5% glycerol, 1 mM DTT, pH 7.5. To separate $NTD_{1-155}$ from its tags, the pure fusion proteins were then incubated with $His_6$-tagged TEV protease for 16 hours at 4 °C by dialysis against a buffer containing 20 mM sodium phosphate, 100 mM NaCl, and 0.5 mM EDTA, pH 8.0. The product of proteolytic cleavage was separated from the tags by $Ni^{2+}$ affinity chromatography, employing a buffer containing 8 M urea (500 mM imidazole, 50 mM sodium phosphate, 100 mM NaCl, 8 M urea, pH 8.0) to prevent the aggregation of the cleaved AR in the column. Finally, the cleaved proteins were stored at −80 °C.

Full length, human Hsp70 (HSPA1A) and Hsp40 (DnaJA2) constructs were transformed into BL21(DE3) cells and single colonies were used to inoculate Terrific Broth (TB) medium containing ampicillin (50 µg mL$^{-1}$). Cultures were grown at 37 °C for 5 h, cooled to 20 °C and induced overnight with 200 µM IPTG. DnaJA2 and Hsp70 (SBD) (residues 394–540) were purified using a $Ni^{2+}$-NTA column, followed by overnight TEV cleavage of the His tag and gel filtration on a Superdex S200 (GE Healthcare). Hsp70 was purified using a $Ni^{2+}$-NTA column, followed by overnight TEV cleavage and subjection to an ATP agarose column. Hsp70 was eluted with ATP and dialyzed to remove excess nucleotide. This purification method provides Hsp70 in its ADP-bound state, based on partial proteolysis. A truncated form of $Hsp70_{SBD}$ (HSPA1A) was used for the FP studies, which encompasses residues 394-509, was purified in a similar way.

**NMR Sample preparation.** The protein solutions stored at −80 °C were thawed and dialyzed for 16 h at 4 °C against a phosphate buffer (sodium phosphate (20 mM), tris(2-carboxyethyl)phosphine (TCEP, 1 mM), pH 7.4). Finally, 10% D$_2$O and 0.015 mM DSS were added to the samples.

**Acquisition and analysis of NMR spectra.** Two-dimensional $^1H$,$^{15}N$-HSQC NMR experiments were recorded at 5 °C on a Bruker 600 or 800 MHz spectrometer equipped with cryogenic probes. Spectra were processed with NMRPipe[66] and analyzed using CcpNMR Analysis[67]. The NMR samples used for the experiments reported in Fig. 1, Supplementary Figs. 1 and 5 contained 33 µM $^{15}N$ single-labelled $NTD_{1-155}$ protein and either 16.5 µM or 33 µM unlabeled Hsp70 and Hsp40 in 20 mM phosphate buffer, pH 7.4, 1 mM TCEP, and 10% (v/v) D$_2$O. The NMR samples used for the experiments shown in Supplementary Fig. 1 were equivalent to those used for the experiments shown in Fig. 1, but contained $NTD_{144-450}$ instead of $NTD_{1-155}$.

The NMR samples used for the experiments reported in Fig. 4a contained 125 and 450 µM $^{15}N$ single-labelled $NTD_{1-155}$ constructs and those used for the experiments reported in Fig. 4b contained 125 µM $^{15}N$ single-labelled $NTD_{1-155}$ constructs and 125 µM unlabeled Hsp70. The $^{15}N$ $R_2$ rates of $NTD_{1-155}$ were

measured with 10 different relaxation delays (t) and fit to the equation $I = I_o \exp(-R_2 t)$ and only the values of $R_2$ determined with less than 5% error were considered sufficiently reliable for the analysis shown in Fig. 4a. The ratio of signal intensities shown in Fig. 4b corresponds to the total signal intensity of the signals that are not affected by chaperone binding (Fig. 1) measured at a given time divided by that measured after 1 day of equilibration.

**Fluorescence polarization assays.** For competition FP studies, 2 µM $Hsp70_{SBD}$ and 25 nM FAM-LVEALY (Anaspec) were incubated with varying concentrations of AR protein for 30 min at room temperature in assay buffer (100 mM Tris, 20 mM KCl, 6 mM MgCl$_2$ pH 7.4). After incubation, fluorescence polarization was measured (excitation: 485 nm emission: 535 nm) using a SpectraMax M5 plate reader. In the case of the FAM-NRLLLTG tracers, 12 µM $Hsp70_{SDB}$, and 25 nM tracer were used and experiment was performed in a similar manner. These experiments were repeated in duplicate and repeated on three independent days.

**Isothermal Titration Calorimetry.** The $NTD_{1-155}$ proteins, as well as $Hsp70_{SBD}$ or Hsp40, were dialyzed overnight against ITC buffer (25 mM HEPES, 5 mM MgCl$_2$, 10 mM KCl pH 7.5). Concentrations were determined using a BCA Assay (Thermo Fisher Scientific), and the experiment was performed with a MicroCal ITC$_{200}$ (GE Healthcare) at 25 °C. AR protein (100–200 µM) in the syringe was titrated into a 10–20 µM cell solution of Hsp70 (SBD) protein. Calorimetric parameters were calculated using Origin 7.0 software and fit with a one or two-site binding model. Experiments were performed in duplicate and repeated twice on two independent days.

**Cell culture and transfection.** Human embryonic kidney 293 T cells (HEK293T cells) were obtained from ATCC (CRL-3216) and maintained in DMEM containing 4.5 g L$^{-1}$ glucose (Glutamax, Gibco) supplemented with 10% (v/v) charcoal stripped calf serum (Gibco) and antibiotics. Cells were cultured in a humidified atmosphere containing 5% CO$_2$ at 37 °C. Transient transfection of HEK293T cells was performed with polyethylenimine (PEI) (Polysciences) at a ratio of 1 µg DNA to 3 µl PEI to express androgen receptor tagged with the FLAG epitope (FLAG-AR), the mutated version (FLAG-mut) and the version lacking the region of sequence targeted by Hsp40 and Hsp70 in the NMR experiments (FLAG-del).

PC12 cells expressing tetracycline-inducible AR112Q (gift of Diane Merry, Thomas Jefferson University) were characterized previously[57]. Cells were maintained in phenol red-free DMEM (21063-029, Invitrogen) supplemented with 5% charcoal stripped fetal calf serum (Thermo Scientific Hyclone Products), 10% charcoal stripped horse serum (Invitrogen), G418 (Gibco) and Hygromycin B (Invitrogen). When indicated, cells were transfected using Lipofectamine 2000 (Life Technologies) or the Neon transfection system (Thermo Scientific) to express human CHIP and HA-His-ubiquitin (gifts from Dr. Yoichi Osawa, Univ. of Michigan). AR expression was induced with 0.5 µg mL$^{-1}$ doxycycline (Clonetech) and activated by treatment with the synthetic androgen R1881. Lactacystin was from Sigma-Aldrich (L6785) and 17-AAG was from LC Laboratories (A-6880).

**Antibodies and reagents.** For Western Blot analyses the following antibodies were used: anti-β-actin-HRP (ab8224, 1:10000) (Abcam), rabbit-anti-FLAG (F7425, 1:2000), and mouse-anti-FLAG M2 (F1804, 1:2000) (Sigma), mouse-anti-HSP70/72 (ADI-SPA-810) and rabbit-anti-HSP40 (ADI-SPA-400-D) (Enzo Life Science).

Additional primary antibodies were used against Akt (C6E7; Cell Signaling Technology), ERK1/2 (9102, Signaling Technology), GAPDH (NB-600-502; Novus Biologicals), HSP90 (H114, sc7947; Santa Cruz Biotechnology), CHIP (PA5-32046, Thermo Scientific, Pierce), HA (MMS-101R, Covance, Biolegend), HSP25 (ADI-SPA-801-488; Enzo Life Science), and HSP70 (ADI-SPA-812; Enzo Life Science). HRP conjugated secondary antibodies were from Bio-Rad (170-6515, 170-6516). For immunofluorescence the following primary antibodies were used: rabbit-anti-FLAG (F7425, 1:200), mouse-anti-FLAG M2 (F1804, 1:200) (Sigma), AlexaFluor conjugated secondary antibodies were from Life Technologies (A11072, 1:300). Mounting medium with DAPI (H-1200, 1:500) was from Vector Labs. For immunoprecipitation, pull down was performed with protein A–agarose beads (Santa Cruz Biotechnology) and the anti-AR antibody (PG21; Millipore).

**Western blotting**. Cells were washed and harvested in PBS, lysed in RIPA buffer (Technova) containing phosphatase and protease inhibitors (Roche). Lysates were centrifuged at 15000 g to separate soluble and pellet fractions. Total protein was quantified using BCA assay (Pierce Biotechnology). Proteins were resolved by 7.5 or 15% SDS-PAGE, transferred to PVDF or nitrocellulose membranes, and probed with specific antibodies.

Filter retardation assay was performed by sample filtration through a 0.2 µm cellulose acetate membrane (Whatman, GE Healthcare) using a slot-blot apparatus.

For ubiquitination studies, cells were homogenized in buffer (0.025 M Tris, 0.15 M NaCl, 0.001 M EDTA, 1% NP-40, 5% glycerol, pH 7.4) containing protease inhibitors and 5 mM N-ethylmaleimide (Sigma-Aldrich). Lysates were incubated on a rotator at 4 °C for 2 h with AR antibody or non-immune rabbit IgG (Santa Cruz Biotechnology). Prewashed protein A–agarose beads were added, and samples were incubated for 1 additional hour at 4 °C. Protein-antibody-bead complexes were washed 6 times in buffer, and proteins were eluted by boiling in SDS-loading buffer for 5 min at 100 °C. Samples were then resolved on by 7.5% SDS-PAGE.

Images of uncropped and unprocessed scans of blots are provided as Source Data files.

Agarose gel electrophoresis for resolving aggregates was performed as previously described[68]. Briefly 150 µg total protein was loaded to 1.5% agarose gels (375 mM Tris buffer, pH 8.6; 0.1% SDS) and electrophoresed at 100 V until the dye front had reached the bottom of the gel. Transfer of proteins was performed using a semi-dry transfer apparatus to PVDF membranes.

**Immunofluorescence**. For immunofluorescence, cells were seeded on sterilized 12 mm microscope coverslips, fixed 15 min in 4% paraformaldehyde at room temperature and permeabilized in ice-cold methanol for 10 min. Cells were then washed, blocked with PBS containing 2% bovine serum albumin for one hour at room temperature and incubated with the corresponding primary antibody diluted in blocking solution for one hour. After washing with PBS, samples were further incubated with the corresponding secondary antibodies for one hour. DAPI was used to label DNA and the coverslips were washed, air-dried and mounted on slides using Prolong Gold antifade mountant (Life technologies). 6 µm cryosections of muscle were fixed in methanol. Slides were stained, mounted, and imaged by confocal microscopy (Olympus FV 500 or Nikon A1). Nuclear signal intensity was quantified by CellProfiler.

**Proximity ligation assay**. The Duolink in situ Proximity ligation (PLA) kit (Olink Bioscience) was used for the detection of protein–protein interactions according to the manufacturer's instructions[45]. The assay is based on the simultaneous binding of two antibodies to the proteins of interest and makes use of a proximity ligation technique combined with an amplification reaction to detect the antibodies whenever they are less than 30 nm apart. HEK293T cells were grown and transfected as indicated on 12 mm microscope coverslips (Thermo). Cells were fixed in PBS containing 4% paraformaldehyde for 15 min and permeabilized with methanol for 5 min. Fixed cells were blocked in PBS containing 2% BSA (Sigma) for an hour and then incubated with the indicated antibodies against Hsp70 (ADI-SPA-810, 1:100), Hsp40 (ADI-SPA-400-D, 1:100) (Enzo Life Science) and FLAG (M2, F1804 and F7425, both 1:50) (Sigma). Samples were subsequently incubated for 1 h at 37 °C with secondary antibody PLA with and without probes for mouse and rabbit (Sigma), and then for 30 min at 37 °C with a ligation solution to ligate the two PLA probes and form a circular DNA template. A rolling-circle amplification was then performed by incubating the cells for 100 min at 37 °C with the amplification solution. The amplicons generated are recognized by fluorescent oligonucleotides present in the amplification solution. Nuclei were stained with DAPI and the coverslips were washed, air-dried and mounted on slides using Prolong antifade mountant (Life Technologies). Cells were analyzed with a ×63 objective on a Leica SP5 confocal microscope. Images were presented as maximal projection of z-stacks by using Fiji.

**AR113Q mice**. AR113Q knock-in mice were generated using exon 1-specific gene targeting[50,59], backcrossed to C56BL/6 (≥10 generations), and housed in a specific pathogen-free facility on a 12-h light/dark cycle with chow and water ad libitum. For analysis, AR113Q males were used at 3 months of age and randomly assigned to treatment groups. JG-294 was dissolved in 70% propylene glycol (W294004, Sigma-Aldrich) and 30% PBS (pH 7.4, Life Technologies), heated to 95 °C to dissolve compound, aliquoted and frozen. Mice received intraperitoneal injections (3 mg kg$^{-1}$) every other day for 2 weeks. 17-DMAG (S1142, Selleckchem) was dissolved in 1% DMSO (D2650, Sigma-Aldrich), 30% polyethylene glycol (202371; Sigma-Aldrich) and 1% Tween80 (P4780; Sigma-Aldrich), and administered by i.p. injection (10 mg kg$^{-1}$) every other day for 2 weeks. At the end of treatment, mice were euthanized and quadriceps were harvested and flash frozen in liquid nitrogen or mounted on OCT (Tissue-Plus, Fisher Health Care) for cryosectioning. For western blot, lysates were obtained by homogenizing tissue in RIPA containing phosphatase and protease inhibitors. All procedures involving mice were approved by the University of Michigan Committee on Use and Care of Animals (PRO00008133) and conducted in accordance with institutional and federal ethical guidelines for animal testing and research.

**Quantitative RT-PCR**. Total RNA was extracted from mouse quadriceps muscles using Trizol (Sigma-Aldrich). cDNA was synthesized from extracted RNA using the High Capacity cDNA Reverse Transcription Kit (Applied Bio Systems). Quantitative PCR was carried out on a 7500 Real-Time PCR System (Applied Biosystems) using FastStart TaqMan Probe Master Mix (Roche) and gene specific primers for AR (universal probe library #58, Roche; 5′ primer: ccagtcccaattgtgtcaaa; 3′ primer: tccctggtactgtccaaacg) and Cpsf2 (Mm00489754_m1, Applied Biosystems). Relative AR expression was determined by the standard curve method and normalized to Cpsf2.

**Statistics**. Statistical significance was determined by analyzing data sets using unpaired Student's $t$-test, one-way ANOVA with Tukey's multiple comparison test or two-way ANOVA with Bonferroni or Niewman–Keuls multiple comparison test in Prism6 (GraphPad). Differences between mean values were defined as significant at $p < 0.05$.

**Reporting summary**. Further information on research design is available in the Nature Research Reporting Summary linked to this article.

## Data availability
The source data underlying the quantifications shown in Figs. 2A, 3B, 4B, 5A, 5D, 6B, 6C, 7B, 7C, 7E, and Supplementary Figs. 3A, 7F, and 9A are available as Source Data files and any other data is available from the corresponding authors upon reasonable request.

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

## Acknowledgements

X.S. acknowledges funding from Obra Social "la Caixa", AGAUR (2017 SGR 324), Marató TV3 (102030), MINECO (BIO2012-31043 and BIO2015-70092-R) and the European Research Council (CONCERT, contract number 648201). IRB Barcelona is the recipient of a Severo Ochoa Award of Excellence from MINECO (Government of Spain). A.P.L. acknowledges funding from NIH (NS055746). J.E.G. acknowledges funding from NIH (NS059690). S.R.N. was supported by a Rackham Predoctoral Fellowship and by the NIH (GM007863).

## Author contributions

B.E., M.M., J.G, and X.S. carried out and analyzed NMR experiments. G.C. set up a protocol to express, purify, and study the aggregation of $NTD_{1-155}$. J.N.R., D.M.C.S., and H.S. carried out FP and ITC experiments. P.M-C. and C.D.S. carried out PLA, WB, and IF experiments. V.C.B., E.G., and S.R.N. analyzed effects of Hsp70-targeted small molecules in cell culture. V.C.B. and Z.Y. analyzed effects of Hsp70-targeted molecules in mice. R.P., I.C.F., I.B-H. and A.N. contributed to the analysis and interpretation of the results. J.E.G., A.P.L. and X.S. conceived and led the project and also wrote the first draft of the paper. All authors contributed to the final version.

## Additional information

**Competing interests:** The authors declare no competing interests.

