## [Peer Review File · Nature Communications]

Reviewers' comments:

Reviewer #1 (Remarks to the Author):

The manuscript submitted by Eftekharzadeh et al. describes studies that support a model for the action of Hsp40 and Hsp70 on the androgen receptor (AR) in which chaperone binding to the intrinsically disordered N-terminal domain (NTD) of AR disrupts an intramolecular interaction between this domain and the ligand-binding domain (LBD) and also suppresses the aggregation propensity of the NTD (including a polyQ region) of the NTD. Previous literature results propose that ligands of the androgen receptor favor exposure of a hydrophobic site on the LBD that is the binding site for the primary motif for chaperone binding. Together with the results reported in this manuscript, which does not report direct binding studies with full-length AR but only with the NTD, the idea of competition between intramolecular binding favored by ligands and chaperone binding is proposed.

This is a very clearly written paper with well-designed experiments that go from biophysical demonstrations of binding using NMR, ITC and fluorescence anisotropy competition binding, to cellular studies of protein-protein interactions, and lastly to mouse models. In the latter two experiments, known inhibitors and modulators of Hsp70 action are used to test the authors' hypotheses. The story presented is compelling and significant. However, the authors have a number of issues that should be addressed before this work is publishable. Most relate to descriptions of methods, which are inadequate in several cases. And in a couple of cases, improvements in clarity of this somewhat complicated story are recommended for both text and figures. Specific points are given below:

1. No where in the paper is it stated when nucleotides are added, and if they are, which ones. This is crucial to understanding the binding assays. The assumption would be that at all times the chaperone added is nucleotide free? This may not alter the conclusions, but must be stated. Moreover, extrapolation to their model for competition between ligand binding and chaperone binding in the cell would be greatly affected by cellular levels of ATP. And one might predict greater aggregation in the presence of ATP. Here again, it appears their assays were done in the absence of nucleotide.
2. The claim that the NMR behavior they see is "slow exchange" or "intermediate exchange" should be supported more convincingly. It is particularly important to point out that they are working at low temperature, which definitely favors slow exchange. It seems likely that they are at slow exchange AND that there are multiple binding modes, otherwise one might expect to see a new set of signals arise from a bound state.

3. It would be helpful to the reader to add a descriptor when acronyms for compounds are used, for example: the ligand analogue R1881, or the inhibitor JG294, or whatever.
4. In Figure 4B, the concentration of NTC is given as 33 micromolar, but in the text it is stated to be 125 micromolar.
5. The colors in the NMR figure (Fig. 1) and in the immunofluorescence figure 6 are not clearly defined. The spectra in the small panels in Fig. 1 are very hard to see. Perhaps it would be better to show slices.
6. The relationship of Hsp40 binding to the entire model is not made explicit.
7. As the Discussion describes the overall model, the reader would be greatly helped by including a schematic such as the one in the graphical abstract.
8. Methods: Not only in the Methods section itself, but in the text description of Results, the authors must describe more fully what they did. A glaring example is the proximity ligation assay using Duolink II PLA probes. I had to go to the Sigma site and hunt down how this method works. Then I realized it is not as straightforward as it seems. Please give details. In another section on preparation of labeled protein, there is no mention of the actual introduction of N15. Also define TB medium.

Reviewer #2 (Remarks to the Author):

Eftekharzadeh et al. have used NMR approaches to study Hsp40 and Hsp70 interactions with the unstructured N-terminal domain of the androgen receptor (AR). This is an interesting yet poorly understood region of AR, and one that is central to AR adopting the agonist conformation pertinent to transcription.

The NMR and binding analyses make a strong case that Hsp40 and Hsp70 contact multiple hydrophobic motifs in the N-terminal domain of AR. The conclusion that the FQNLF is the preferred site of binding fits the narrative, but the data for this is not convincing, and one wonders if most any protein fragment with hydrophobic motifs would behave similarly in the assays. Because deleting the FQNLF affects multiple aspects of AR protein behavior, the experiment in Fig. 3 using PLA is inadequate to draw the conclusion that this is the primary site of chaperone contact with AR.

The small molecule experiments designed to test for AR polyubiquitination (Fig. 5A, B) in response to Hsp70 inhibition seem to generate small effects at near-background levels on the blots. At a minimum these experiments require validation by siRNA approaches.

The question of protein clearance of AR113Q in response to small molecules should be done in a manner that determines the half-life of soluble versus insoluble pools.

The topic is interesting but the study may be better suited for a biochemical journal. The molecular link between the FQNLF in the AR N-terminal domain and Hsp40/Hsp90 by NMR is convincing, but the linkage in cells is not compelling. For this reason, the study seems preliminary.

Reviewer #3 (Remarks to the Author):

Using a variety of approaches, Eftekharzadeh et al. examine the interaction of Hsp70 and Hsp40 with the androgen receptor (AR). Data support a model whereby Hsp70 and Hsp40 interact with the same five residue motif in the AR N-terminus that is critical for the AR N-terminus to interact with its C-terminal ligand binding domain during transcriptional activation upon ligand binding to AR. The AR-Hsp70 interaction maintains AR solubility and decreases aggregation of AR with an expanded polyQ. Importantly, using a small molecule to stabilize the AR-Hsp70 complex in vivo promotes degradation of AR. Overall this is a very nice study that not only furthers our understanding of the inter and intra molecular interactions that are involved in AR biology, it also provides a novel therapeutic pathway towards treating patients affected with SBMA. Minor suggestions for improving presentation of the work include:

1. On page 7, middle of first paragraph NMR data are described using the AF-1 region of AR and Hsp70/Hsp40. The authors state that little change in the spectrum was indicating little to no interaction. The spectra data should be presented as supplemental material.

2. Page 15, bottom paragraph data in Figure 5C that JG-series small molecules effectively increase AR clearance in absence of inducing a stress response. Since this point is very critical to the therapeutic potential of these compounds the author need to expand this section describing those cellular proteins whose expression was assessed re evaluating the stress response. Particular emphasis should be place on a description as to why no change in their expression is strong proof for the lack of a stress response.

Reviewer 1

The manuscript submitted by Eftekharzadeh et al. describes studies that support a model for the action of Hsp40 and Hsp70 on the androgen receptor (AR) in which chaperone binding to the intrinsically disordered N-terminal domain (NTD) of AR disrupts an intramolecular interaction between this domain and the ligand-binding domain (LBD) and also suppresses the aggregation propensity of the NTD (including a polyQ region) of the NTD. Previous literature results propose that ligands of the androgen receptor favor exposure of a hydrophobic site on the LBD that is the binding site for the primary motif for chaperone binding. Together with the results reported in this manuscript, which does not report direct binding studies with full-length AR but only with the NTD, the idea of competition between intramolecular binding favored by ligands and chaperone binding is proposed.

This is a very clearly written paper with well-designed experiments that go from biophysical demonstrations of binding using NMR, ITC and fluorescence anisotropy competition binding, to cellular studies of protein-protein interactions, and lastly to mouse models. In the latter two experiments, known inhibitors and modulators of Hsp70 action are used to test the authors' hypotheses. The story presented is compelling and significant. However, the authors have a number of issues that should be addressed before this work is publishable. Most relate to descriptions of methods, which are inadequate in several cases. And in a couple of cases, improvements in clarity of this somewhat complicated story are recommended for both text and figures. Specific points are given below:

1. No where in the paper is it stated when nucleotides are added, and if they are, which ones. This is crucial to understanding the binding assays. The assumption would be that at all times the chaperone added is nucleotide free? This may not alter the conclusions, but must be stated. Moreover, extrapolation to their model for competition between ligand binding and chaperone binding in the cell would be greatly affected by cellular levels of ATP. And one might predict greater aggregation in the presence of ATP. Here again, it appears their assays were done in the absence of nucleotide.

We apologize for the omission. Indeed, as suggested by Reviewer 1, Hsp70 is purified in its ADP-bound state and we did not add excess ATP to test the effects of nucleotide. We chose this experimental route to avoid any contribution of ongoing ATP hydrolysis to the observed binding events, which is especially important because of the long timescales of the NMR-based aggregation studies. In some systems, non-hydrolyzable ATP analogs can be used, but, in the specific case of Hsp70, it is not clear that these molecules trap the intended structural state and it is not clear that they would maintain that state on long incubations. In the revised manuscript, we make the nucleotide state clearer, both in the main text and in the corresponding figure captions. In addition, we have added a part to the Discussion, in which we discuss how ATP binding and hydrolysis *in vivo* may influence the way that Hsp70 binds its substrates, using analogies to other systems. We thank Reviewer 1 for raising this important point.

The reviewer also raises an interesting point about how changes in ATP/ADP ratio or levels might contribute to AR dyshomeostasis *in vivo*. The affinity of Hsp70 for its nucleotides is in the low micromolar (or high nanomolar) range, while ATP and ADP

are typically present at millimolar levels in cells. So, any changes in cellular ATP/ADP would need to be dramatic to start impacting chaperone cycling and function. However, oxidative stress is known to inactivate the chaperone (Mol. Cell, 2005; 17:381-392 and Mol. Biol. Cell, 2012; 23:3290, etc), so the reviewer's comment is interesting and worthy of future study.

2. The claim that the NMR behavior they see is "slow exchange" or "intermediate exchange" should be supported more convincingly. It is particularly important to point out that they are working at low temperature, which definitely favors slow exchange. It seems likely that they are at slow exchange AND that there are multiple binding modes, otherwise one might expect to see a new set of signals arise from a bound state.

We agree with Reviewer 1 that it is important to emphasize that the experiments were carried out at low temperature and that this choice contributes to the intermediate-slow exchange behavior that we describe in the manuscript. We were forced to use this temperature because it maximizes the quality of the spectrum of NTD₁₋₁₅₅ as it decreases solvent exchange and have specified this in the revised manuscript. We fully agree with Reviewer 1 that this behavior is likely due to a combination of slow exchange and multiple binding modes, and have added text to reflect this in the main text, together with a relevant reference.

3. It would be helpful to the reader to add a descriptor when acronyms for compounds are used, for example: the ligand analogue R1881, or the inhibitor JG294, or whatever.

Yes, there is a lot of nomenclature here, so it is a good idea to be explicit. In the revised manuscript, we have been more careful to identify each compound and its intended use. For example, R1881 refers to metribolone, a synthetic androgen, while JG-98 and JG 294 are Hsp70 modulators.

4. In Figure 4B, the concentration of NTC is given as 33 micromolar, but in the text it is stated to be 125 micromolar.

The concentration used was 125 μ M. We have corrected this in the revised manuscript and thank Reviewer 1 for highlighting this error.

5. The colors in the NMR figure (Fig. 1) and in the immunofluorescence figure 6 are not clearly defined. The spectra in the small panels in Fig. 1 are very hard to see. Perhaps it would be better to show slices.

We have added definitions to the colors used in Figures 1 and 6 and shown the spectra in the small panels as slices to improve the clarity of the Figure as suggested by the Reviewer.

6. The relationship of Hsp40 binding to the entire model is not made explicit.

We thank Reviewer 1 for highlighting this issue, which we have addressed in multiple ways. First, we have added text in the revised Discussion to explain how Hsp40 assists Hsp70 by interacting with substrates and transferring them to Hsp70 and by

accelerating its catalytic cycle. We have also added ITC data to show that Hsp40 binds with approximately equal (or slightly better) affinity to NTC₁₋₁₅₅ (Fig. S4B). This result helps place Hsp40 into the model, as the initial recruiter of Hsp70.

7. As the Discussion describes the overall model, the reader would be greatly helped by including a schematic such as the one in the graphical abstract.

We have converted the graphical abstract into a new Figure (Fig. 8) to help the readers understand the Discussion and thank the Reviewer for this suggestion.

8. Methods: Not only in the Methods section itself, but in the text description of Results, the authors must describe more fully what they did. A glaring example is the proximity ligation assay using Duolink II PLA probes. I had to go to the Sigma site and hunt down how this method works. Then I realized it is not as straightforward as it seems. Please give details. In another section on preparation of labeled protein, there is no mention of the actual introduction of N15. Also define TB medium.

We have gone through the Online Methods document and, especially, the Results section made changes to ensure that our description of our experimental procedures is sufficiently detailed to allow others to reproduce our results. In particular, we have ensured that is the case for the preparation of the NMR samples reported in Figures 1 and 4 and the PLA experiments reported in Figures 3 and S4. In addition, to address the reviewer's specific point about the PLA studies, we have added text to outline the general approach – as others may not be familiar with this method.

Reviewer 2

Eftekharzadeh et al. have used NMR approaches to study Hsp40 and Hsp70 interactions with the unstructured N-terminal domain of the androgen receptor (AR). This is an interesting yet poorly understood region of AR, and one that is central to AR adopting the agonist conformation pertinent to transcription.

The NMR and binding analyses make a strong case that Hsp40 and Hsp70 contact multiple hydrophobic motifs in the N-terminal domain of AR. The conclusion that the FQNLF is the preferred site of binding fits the narrative, but the data for this is not convincing, and one wonders if most any protein fragment with hydrophobic motifs would behave similarly in the assays. Because deleting the FQNLF affects multiple aspects of AR protein behavior, the experiment in Fig. 3 using PLA is inadequate to draw the conclusion that this is the primary site of chaperone contact with AR.

This is an important point and we have made numerous changes to the manuscript in response.

A. To be clear, the NMR studies suggest that Hsp70 and Hsp40 bind to a region that includes the FQNLF motif, as well as amino acids on either side. We did not mean to suggest that the chaperones only bind to the FQNLF residues. We focused on those residues because of their known biological importance, but the chaperones clearly have interactions on either side. In the revised text and

abstract, we have been careful to use terms such as “the region of the NTD that includes the FNQLF motif”.

- B. We have now conducted binding studies with NTD₁₄₄₋₄₅₀, a disordered stretch of AR that contains multiple hydrophobic motifs. For example, this construct is twice as large as NTD₁₋₁₅₅ and contains at least two motifs with aromatic/hydrophobic residues equivalent to FQNLF: ¹⁸¹LKDIL¹⁸⁵ and ⁴³⁵WHTLF⁴³⁹. Importantly, we did not observe any significant binding of either Hsp70 or Hsp40 to either site in that protein by NMR (Fig. S1). This is a critically important experiment for Reviewer 2’s comment because it shows that the chaperones will not indiscriminately bind to all hydrophobic motifs under these conditions. Rather, in the region of sequence that contains FQNLF, we see that only a third of the signal intensity is left after addition of either chaperone (Fig. 1). To be clear, there are a handful of other, weaker binding sites in NTD₁₋₁₅₅, where we see smaller changes in intensity that are attributed to transient binding. We have also been careful to indicate these secondary sites in the revised manuscript. Finally, the reviewer is correct that tighter binding does not necessarily mean that the region around FQNLF is the “primary site”, so we have been careful to never claim this. Regardless, a preponderance of evidence points to this interaction being an important one.
- C. In addition to deleting the FQNLF motif (FLAG-AR-del; Fig. 3), we have also performed NMR and ITC experiments shown in Figures S2 and S4 where we show that more subtle mutation of the three hydrophobic residues of the FQNLF motif to Ala (AQNAA) leads a 10-fold weaker interaction of NTD₁₋₁₅₅ with either molecular chaperone. This data clearly shows that the FQNLF region is important for overall affinity. Moreover, we show that both the AQNAA mutant (FLAG-AR-mut) and the deletion (FLAG-AR-del) are able to translocate into the nucleus in response to androgen (Fig S4). Thus, manipulating this region does not damage a subset of early AR functions.

The small molecule experiments designed to test for AR polyubiquitination (Fig. 5A, B) in response to Hsp70 inhibition seem to generate small effects at near-background levels on the blots. At a minimum these experiments require validation by siRNA approaches.

Prior work has shown that allosteric regulation of Hsp70 to favor the ADP-bound state has modest effects on client protein ubiquitination. These studies used both genetic approaches (over-expression of the Hsp70 co-chaperone Hip) and small molecule (YM1), and examined ubiquitination of Hsp70 clients, including nNOS and polyQ AR (Nat Chem Biol, 2013; 9:112-8; J Clin Invest, 2015; 125:831-45). The magnitude of the effects shown here using JG98 are in line with those published studies. We note that the approach we’ve taken is to favor Hsp70’s ADP-bound state and slow cycling of Hsp70 with misfolded clients such as polyQ AR. This is distinct from inhibiting Hsp70 function by either small molecules or genetic knockdown, both of which would be expected to have detrimental effects and impair Hsp70-dependent ubiquitination. Nonetheless, we agree with the Reviewer’s comment that it would be beneficial to provide evidence that the effects we’re seeing are mediated by Hsp70. Toward that end, we treated cells expressing polyQ AR with JG-258, a structural analog of JG-98 that is modified so that it does not bind Hsp70 (J Med Chem, 2018: 61:6163-77). In new Figure S9, we demonstrate that treatment with JG-258 does not

change steady state polyQ AR levels in cells nor does it alter polyQ AR ubiquitination. These data support our interpretation that the effects of JG-98 on polyQ AR levels and ubiquitination require interaction with Hsp70.

The question of protein clearance of AR113Q in response to small molecules should be done in a manner that determines the half-life of soluble versus insoluble pools.

To address this we used PC12 cells that express either WT (AR10Q) or polyQ AR (AR112Q) from a tetracycline-responsive promoter to examine effects on protein clearance. AR expression was induced for 48 hrs in the presence of ligand, then doxycycline was removed and we followed AR clearance by western blot for 24-72 hrs. We performed this chase in the presence of active (JG-98) or inactive (JG-258) Hsp70 targeted small molecules, or in the presence of vehicle. Lysates were collected from soluble (supernatant) and insoluble (pellet) fractions and examined by western blot. As shown in new Figure S8, polyQ AR was readily detected in both soluble and insoluble fractions, and showed enhanced clearance from both fractions by JG-98 but not JG-258 or vehicle. In contrast, only a scant amount of AR10Q was present in the insoluble fraction, and clearance of WT AR were not markedly effected by either JG-98 or JG-258.

The topic is interesting but the study may be better suited for a biochemical journal. The molecular link between the FQNLF in the AR N-terminal domain and Hsp40/Hsp90 by NMR is convincing, but the linkage in cells is not compelling. For this reason, the study seems preliminary.

We were glad to read that Reviewer 2 found the topic interesting and trust that (s)he will find that the changes that we have introduced to address her/his comments as well as those of the other two Reviewers have led to a manuscript acceptable for publication in *Nature Communications*.

Reviewer 3

Using a variety of approaches, Eftekharzadeh et al. examine the interaction of Hsp70 and Hsp40 with the androgen receptor (AR). Data support a model whereby Hsp70 and Hsp40 interact with the same five residue motif in the AR N-terminus that is critical for the AR N-terminus to interact with its C-terminal ligand binding domain during transcriptional activation upon ligand binding to AR. The AR-Hsp70 interaction maintains AR solubility and decreases aggregation of AR with an expanded polyQ. Importantly, using a small molecule to stabilize the AR-Hsp70 complex in vivo promotes degradation of AR. Overall this is a very nice study that not only furthers our understanding of the inter and intra molecular interactions that are involved in AR biology, it also provides a novel therapeutic pathway towards treating patients affected with SBMA. Minor suggestions for improving presentation of the work include:

1. On page 7, middle of first paragraph NMR data are described using the AF-1 region of AR and Hsp70/Hsp40. The authors state that little change in the spectrum was indicating little to no interaction. The spectra data should be presented as supplemental material.

We agree. As explained in our response to Reviewer 2, we have added the requested spectra to the supplementary information as Figure S1. This result is important because NTD₁₄₄₋₄₅₀ is large and contains multiple hydrophobic motifs, yet neither Hsp70 nor Hsp40 interact with it.

2. Page 15, bottom paragraph data in Figure 5C that JG-series small molecules effectively increase AR clearance in absence of inducing a stress response. Since this point is very critical to the therapeutic potential of these compounds the author need to expand this section describing those cellular proteins whose expression was assessed re evaluating the stress response. Particular emphasis should be place on a description as to why no change in their expression is strong proof for the lack of a stress response.

The revised text better introduces the idea that Hsp25, Hsp40 and Hsp70 are used as biomarkers for the stress response as their expression is regulated by Hsf-1. We have clarified this at the top of page 16 in the revised manuscript. We thank the reviewer for the comment.

REVIEWERS' COMMENTS:

Reviewer #1 (Remarks to the Author):

The authors have satisfactorily addressed the points raised in my previous review.

Reviewer #2 (Remarks to the Author):

The authors were responsive to the queries and I am satisfied the manuscript is suitable for publication.

Reviewer 1

The authors have satisfactorily addressed the points raised in my previous review.

We were glad to read that all the concerns of Reviewer 1 were satisfactorily addressed in the revised version and thank her/him for her/his help in improving the manuscript.

Reviewer 2

The authors were responsive to the queries and I am satisfied the manuscript is suitable for publication

We were glad to read that Reviewer 2 considers that the manuscript is ready for publication and thank her/him for her/his help in improving it.